# Analysis of the Influence of Socio-Economic Factors on Occupational Safety in the Construction Industry

**Bożena Hoła** and **Tomasz Nowobilski *** 

Department of Construction Technology and Management, Faculty of Civil Engineering, Wroclaw University of Science and Technology, 50-370 Wrocław, Poland
* Correspondence: tomasz.nowobilski@pwr.edu.pl; Tel.: +48-71-320-39-71

**Abstract:** The purpose of the article was to identify the socio-economic factors generated in a construction environment, which affect the number of accidents at a construction site. Moreover, the objective was to construct a mathematical model that correlates selected factors with the number of employees injured in accidents at work at a construction site, and also to estimate the influence of the identified factors on the level of occupational safety. Based on the analysis of the literature, it was stated that there are no studies describing the impact of socio-economic factors on accident rates in construction. The research included 104 factors that characterize the production value, the potential of enterprises, the generic structure of entities, and also employment and its volatility in the construction industry. In order to solve the problem, multiple regression analysis, available in the Statistica software, was applied. The developed mathematical multifactor model reflects empirical values very well, which was confirmed by the values of multiple correlation coefficients, the coefficient of determination, the adjusted coefficient of determination, as well as the mean square error and root mean square error. The construction of the model does not include qualitative factors, e.g., factors that describe the level of safety culture in the society. The developed model was used to determine the number of people injured in accidents at work. The model has certain limitations regarding its applicability. The model was developed for four selected Polish voivodeships that have a similar level of economic development and occupational safety.

**Keywords:** Construction industry; accidentality; accident factor; multifactor mathematical model; multiple regression; statistical data

## 1. Introduction

Construction is one of the most important sectors of each country's economy. Construction works include activities such as the erection of complete building structures as well as works related to the reconstruction, extension, rebuilding, renovation, modernization, maintenance and use of permanent and temporary building structures [1]. Such a large variety of construction works, and also the impact of changing implementation conditions on their course, creates a large number of hazards to employees, which in turn may lead to occupational accidents or so-called potentially accidental events. This is confirmed by, among others, the previous results of surveys of statistical data [2–4]. For example, the fatality rate in the Polish construction industry in 2016 was equal to 4.3 people per every 100,000 employees; in Estonia, 16.8 people per every 100,000 employees; in Sweden, 2.6 people per every 100,000 employees [5]. The causes of high accident rates include, among others, a low culture of occupational safety [6]. Although accident rates have been declining in recent years, they are still too high and unacceptable [3,5].

Occupational accidents and potentially accidental events are undesirable phenomena because they generate material and moral damage, and also adversely affect economic development [7]. Therefore,

the identification of hazards that may occur at a construction site is an important element of accident prevention [8], the aim of which is to achieve the ideal state of zero occupational accidents.

In the case of issues related to occupational safety, researchers most often focus on the analysis of direct causes leading to occupational accidents. It should be noted, however, that each construction site is located in a specific environment, which may indirectly affect the accident rate in this industry. For example, decisions taken in the sphere of politics and economic management have an impact on the development of the construction sector. A high correlation was found between the value of production and the number of accidents [9,10]. It was also noted [11] that the level of occupational safety is strongly influenced by human factors, such as leadership and management skills, experience and knowledge of employees, level of training, mutual communication, as well as factors that motivate the execution of safe work, e.g., prizes, and also other forms of recognition.

The literature survey indicated that there are no scientific studies regarding the impact of socio-economic factors on the number of accidents in building construction.

This observation provided the impulse to undertake such research and to fulfill the stated research gap. The purpose of the research described in this article is as follows:

1. The identification of socio-economic factors generated in the construction environment;
2. The formulation of numerical indicators that describe the identified factors;
3. The construction of a mathematical model that combines the selected factors, enables their impact on occupational safety to be assessed, and predicts the number of people injured in occupational accidents in the construction industry.

Multiple regression analysis and Statistica software were used to solve the task.

The article consists of the following chapters: Section 2 consists of a literature survey and a justification for undertaking the research topic. Section 3 presents the proposed research methodology in detail. Section 4 contains an example that illustrates the use of the developed methodology, as well as an analysis of the impact of selected socio-economic factors on occupational safety. Finally, Section 5 contains conclusions from the conducted research.

## 2. Literature Survey

The number of accidents at construction site is affected by a number of factors related not only to the construction site but also to the environment. In the analyses concerning the state of occupational safety, the incidental factor should be understood as all kinds of material and intangible actions that have a direct or indirect impact on the accidentality phenomenon [10].

The identification and analysis of accident factors is an important area of scientific research, as it is the basis for the proper design and planning of training and preventive actions, which in turn leads to reduced accidents at work [10,12–17].

The authors of the studies published in [12] emphasize that all participants of a project, as well as the environmental conditions in which the construction is carried out, are responsible for occupational accidents. In papers [13,14], a significant number of accident factors were identified, which were then classified into five groups related to: clients, consultants, contractors, construction workers and construction sites. Among the identified factors, the most important factors were deficiencies in the personal equipment of employees, a defective technical condition of devices, a lack of training, and also inappropriate use of building materials and devices.

Choudhry and Fang [15] presented the results of surveys, conducted directly among construction workers, on the identification of accident factors. The identified factors were assigned to groups related to management, safety procedures, psychological characteristics of employees, economics and work efficiency, self-assessment, experience and perception of risk by employees, the work environment, training and education.

In paper [10], a research questionnaire was used to identify accident factors. The identified accident factors were divided into three groups depending on their degree of connection with an

accident. Group I consisted of factors directly related to a construction site, group II included factors generated in an organization, a construction enterprise, and group III consisted of factors generated in the environment of a construction enterprise.

The paper by Stepien [16] deserves special attention among studies on this subject matter. The author, based on an analysis of the subject literature, made a detailed classification of accident factors, and then divided them into four main groups related to: working conditions, employees, management and organization of work, as well as work environment. Moreover, studies [17] in which the authors proposed a methodology for the classification of direct causes of accidents on scaffolding are also interesting. Using Pareto–Lorenzo analysis, the authors of the research identified causes that have the greatest impact on the occurrence of accidents.

The phenomenon of accidentality is complicated, complex and multi-faceted. In order to thoroughly examine various aspects of this phenomenon, models that reflect selected features are developed. The modeling of the phenomenon of accidentality in building construction has involved the use of, among others, linear mathematical models, descriptive and graphic models, as well as IT models, e.g., neural networks.

The research presented in papers [18,19] included qualitative and quantitative analyses, the effect of which was the creation of mathematical models of the development of the accident trend in the Polish construction industry, as well as cause and effect models that refer to the relationship between the number of people injured in occupational accidents and the identified causes of accidents. The models proposed in the studies showed a very high degree of linear mapping of the analyzed relationships.

The high linear dependence between factors that characterize construction production and the number of accidents in the construction industry was also demonstrated by the authors' previous studies [10,20]. The factors included in the analyses were related to the achieved value of construction production, as well as the type of investment and the nature of the conducted construction works.

In paper [21], classification models were used to identify hazards at a construction site. In turn, research conducted in Korea [22] aimed to develop a model that describes the relationship between risk factors related to the work environment and employees, and the consequences of an accident in the construction industry. The conducted statistical analysis enabled conclusions to be formulated, which showed that the roles of managers and safety managers are crucial for the reduction of hazard in the construction industry. With appropriately conducted preventive and training activities, the risk and consequences of accidents are significantly reduced.

Researchers from Thailand and Malaysia [23] proposed a multi-level model of intervention in the field of safety. The data used to develop the model was obtained from surveys.

Neural models are also used to solve problems related to occupational safety in construction practice. For example, Indian scientists proposed the use of artificial neural networks (ANN) to predict the safe behavior of construction workers [24,25]. In turn, in American studies [26], the authors proposed an approach based on the Bayesian network (BN) in order to diagnose the risk of falling from heights. As a result of the proposed approach, it is possible to determine the probability of an accident, which is associated with various states of hazards regarding safety.

Due to the development of IT technology, virtual modeling of the construction environment and the hazards that occur in it is possible. Therefore, paper [27] presents a method of predicting and monitoring hazards, which is based on a virtual model of a construction site. Research [28] proposed the use of a model of a construction site, which was developed in a virtual environment, for training and educational purposes. In turn, paper [29] proposed the use of virtual reality to identify hazards, as well as to determine the rules of movement of employees in this environment and the impacts of the construction environment on its surroundings. In turn, in research [30], the authors proposed a model that allows construction workers to be tracked in real time using real-time locating systems (RTLS). The developed tool can be used to control the exposure of workers in the construction industry to hazards that change over time.

In order to improve occupational safety, the authors of research [31] proposed an approach based on machine learning (ML). According to the authors, all accidents run according to well-defined patterns, which give the opportunity to predict them. The ML technique has been used to predict accidents related to slip, trip, fall (STF) hazards. The developed model for predicting STF type accidents used the particle swarm optimization (PSO) algorithm.

In turn, a methodology of investigating the phenomenon of accidentality, which is understood as a sequence of consecutive events, was proposed in [32]. A set of 485 occupational accidents that took place between 2008 and 2016 in various circumstances was used to carry out the analyses. The research, conducted using the constructed IT model, confirmed the results of the studies included in paper [31], namely that accidents in the construction industry most frequently occur as a result of slips, trips and falls.

Based on the conducted subject literature survey, it has been stated that the vast majority of research concerns the identification of accident factors generated at a construction site, the direct sources of which are the elements of the conducted construction process, including employees. In turn, it was also observed [20] that the surroundings of a construction site, as well as decisions taken there, may affect the level of occupational safety. This is confirmed by, among others, research [33], in which attention was drawn to the impact of the social environment on the culture of occupational safety.

The conducted subject literature analysis confirms the lack of research of accident factors generated in the external environment of a construction site. The consequence of this is also the lack of models describing the relationship between the identified factors and the number of accidents in the construction industry. The authors of this article have tried to fill this identified research gap. They drew attention to selected socio-economic factors that describe the volume of construction production, the number and structure of construction companies, capital expenditure incurred in the construction industry and employment, as well as their impact on occupational safety.

## 3. Proposed Research Methodology

The external environment of a construction site is assumed by the authors of the article to be the contractual sphere of making decisions that affect the achieved effects of construction activity.

Based on the authors' previous research [34,35], it was found that the basic factors that characterize construction activity are: the value of construction production, the potential of construction enterprises, the generic structure of business entities, employment and its volatility, and also occupational safety. Each of the above-mentioned factors is characterized by a number of sub-factors, which describe in detail the examined aspect. For example, in order to fully characterize the value of construction production, it is required to possess detailed information on the financial outlays spent on construction activity with regards to the types of construction works (investment and renovation works), types of erected construction objects, the size of construction entities carrying out the construction, and others.

As a result of the analysis of published statistical data [36], 104 sub-factors, which describe various aspects of the economic situation in Polish, were identified. Each of the identified sub-factors is described by the numerical index $I_n$ that expresses, e.g., the value of construction production, the number of people employed in the construction industry, etc.

The purpose of the calculations was to develop a model of the accidentality phenomenon, which takes into account the impact of significant socio-economic sub-factors on the number of accidents in the construction industry. The following statistics were used as the indicators of the quality of the model's adjustment to empirical values: multiple correlation coefficient $R_P$, coefficient of determination $R^2$, corrected coefficient of determination $R_S^2$, and also the following measures of volatility: Mean Squared Error (MSE) and Root Mean Squared Error (RMSE). The proposed methodology for modeling the phenomenon of accidentality is shown in Figure 1.

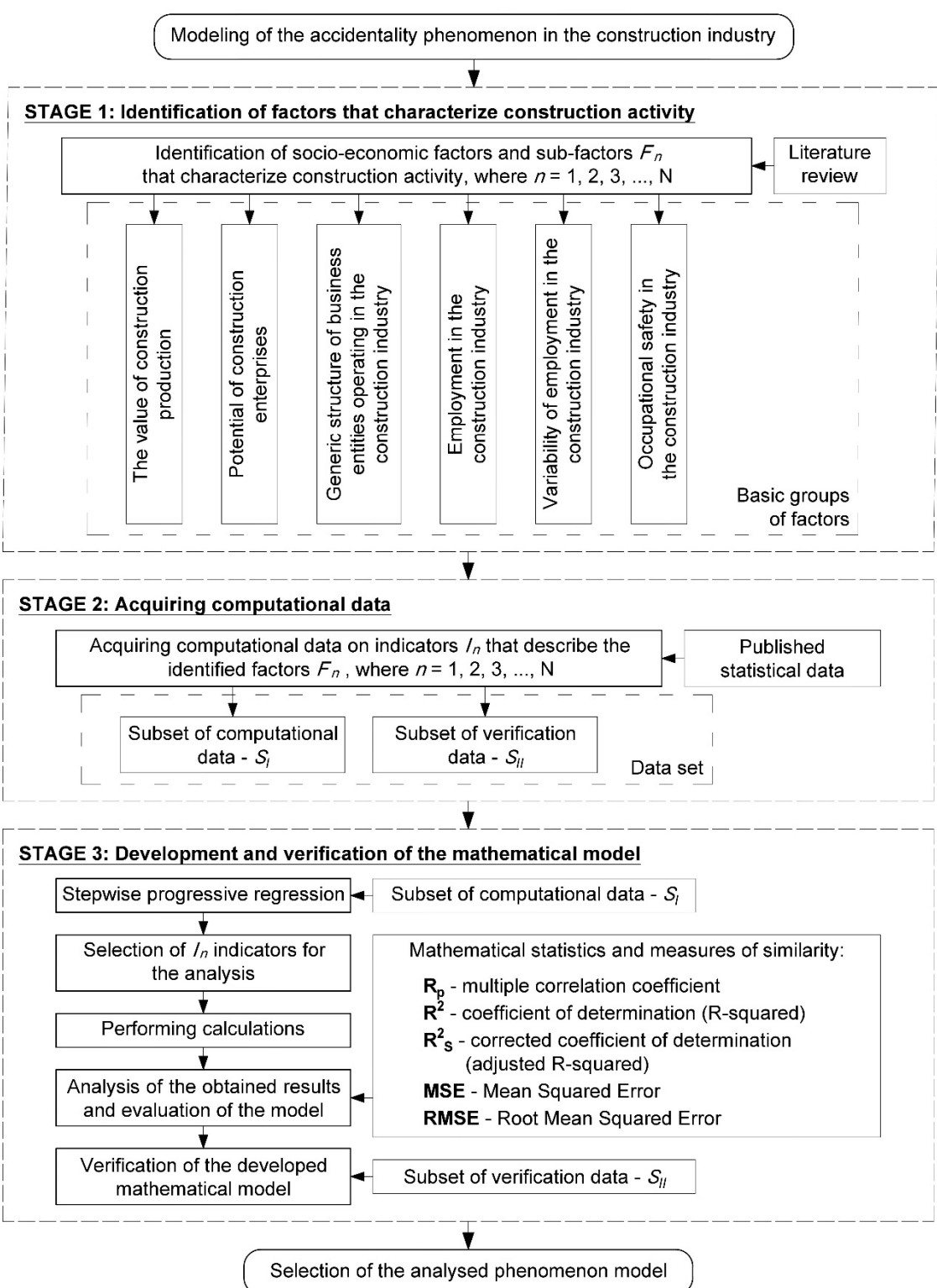

**Figure 1.** Developed methodology for modeling the phenomenon of accidentality in the construction industry.

## 4. Results and Discussion

### 4.1. Multifactor Mathematical Model of the Accidentality Phenomenon in the Construction Industry

The authors used the proposed methodology in order to develop a mathematical linear regression model that describes the accidentality phenomenon in four Polish voivodships: Dolnośląskie, Pomorskie, Małopolskie and Wielkopolskie. The basis for the selection of the above voivodships were the results of the research published in paper [34], which showed that these voivodships are similar in terms of the pace of development and the level of occupational safety in construction.

Input data that characterize these voivodships and cover the period from 2008 to 2017 was obtained from statistical yearbooks published by the Central Statistical Office [36]. A fragment of the data used for the analysis of the Dolnoslaskie voivodeship is presented in Table 1.

**Table 1.** Exemplary values of selected indicators for the Dolnoslaskie Voivodeship in the construction industry [36].

| No. $n$ | Numeric indicator $I_n$ | Designation | Unit | Year 2008 | 2009 | ... | 2017 |
|---|---|---|---|---|---|---|---|
| 1 | The value of construction and assembly production associated with investments in housing | $V_1$ | Million PLN | Lack of data | 840.5 | ... | 625.7 |
| ... | ... | ... | ... | ... | ... | ... | ... |
| 21 | The value of current assets of construction enterprises associated with finished products | $P_1$ | Million PLN | 72.7 | 138.6 | ... | 54.4 |
| ... | ... | ... | ... | ... | ... | ... | ... |
| 42 | The number of entities registered as state-owned enterprises | $S_1$ | Number of entities | 4 | 4 | ... | 4 |
| ... | ... | ... | ... | ... | ... | ... | ... |
| 49 | The number of working women | $E_1$ | Number of people | Lack of data | 7387 | ... | 7636 |
| ... | ... | ... | ... | ... | ... | ... | ... |
| 60 | The number of employed people | $EV_1$ | Number of people | 14,096 | 10,723 | ... | 8397 |
| ... | ... | ... | ... | ... | ... | ... | ... |
| 104 | The number of people injured in occupational accidents | $A$ | Number of people | 892 | 720 | ... | 377 |

$A$ —the number of people injured in occupational accidents, $V_1$—the value of construction and assembly production associated with investments in housing, $P_1$—the value of current assets of construction enterprises associated with finished products, $S_1$—the number of entities registered as state-owned enterprises, $E_1$—the number of women working, and $EV_1$—the number of people employed.

All data was divided into two subsets: 80% was associated with a subset of computational data $S_I$ that was used to create the model, while the remaining 20% was associated with $S_{II}$. data that was used for its verification. After conducting calculations on the set $S_I$ using the "multiple regression" module, the following equation of multiple regression was obtained:

$$A = 266.79 - 0.12 \cdot S_2 - 0.59 \cdot P_1 + 0.03 \cdot S_3 + 0.02 \cdot P_2 - 0.06 \cdot E_1 + 0.73 \cdot EV_2 + 0.20 \cdot V_1 - 39.53 \cdot S_1 \\ + 0.03 \cdot EV_1 + 1.30 \cdot P_4 - 0.11 \cdot P_3 - 0.18 \cdot E_2 - 1.19 \cdot P_5 \tag{1}$$

where:

$A$—the number of people injured in occupational accidents,
$V_1$—the value of construction and assembly production associated with investments in housing,
$P_1$—the value of current assets of construction enterprises associated with finished products,
$P_2$—the capital of enterprises,

$P_3$—investment expenditure in the construction industry,

$P_4$—investment expenditure for machines, technical devices and tools,

$P_5$—investment expenditure for means of transport,

$S_1$—the number of entities registered as state-owned enterprises,

$S_2$—the number of entities registered as commercial companies,

$S_3$—the number of entities run by self-employed people,

$E_1$—the number of women working,

$E_2$—the number of owners and co-owners of enterprises, as well as family members who help free of charge,

$EV_1$—the number of people employed, and

$EV_2$—the number of employed people who have returned from parental leave.

The number and type of explanatory variables that were used in the linear regression model were selected using the F (Fisher–Snedecor) test. In the calculations carried out, the threshold value of the F test that enables another variable to be introduced into the model took the values from 3 to 4, which are recommended by [37]. This corresponded to approximately 5% of the significance level of the F test ($\alpha = 0.05$).

In order to examine the quality of matching the developed mathematical model to the empirical values, the values of the statistics, $R_P$, $R^2$, $R_S^2$, and measures of variability, MSE and RMSE, were determined and are presented in Table 2.

**Table 2.** Summary of parameters that characterize the developed mathematical model.

| No. | Name of the characterizing parameter | Value |
|:---:|:---:|:---:|
| 1 | Multiple correlation coefficient, $R_P$ | 0.997 |
| 2 | Coefficient of determination, $R^2$ | 0.993 |
| 3 | Adjusted coefficient of determination, $R_S^2$ | 0.988 |
| 4 | Mean square error, MSE | 440 |
| 5 | Root mean square error, RMSE | 21 |

When analyzing the obtained results, it should be stated that the developed model is characterized by a very high adaptation to reality (the obtained statistics: $R_P$, $R^2$ and $R_S^2$ are close to one).

In the considered issue, the multiple correlation coefficient $R_P$, which is a measure of the linear relationship between the number of people injured in occupational accidents in the construction industry (A) and the linear combination of selected indicators ($I_n$), reached the value of $R_P = 0.997$. This indicates a very good adjustment of the model to reality [38].

The coefficient of determination $R^2$ provides information on what part of the variation of the explanatory variable A was determined by the model. In the analyzed case, it reached the value of $R^2 = 0.993$, which means over 99% of efficiency when determining the variability of explanatory variable A [39].

The adjusted coefficient of determination $R_S^2$ is a measure of the quality of the model's adjustment, which is dependent on the number of explanatory variables ($I_n$). The obtained value of $R_S^2 = 0.988$ indicates that the appropriate number and types of explanatory variables are included in the model.

The RMSE contains information about the difference between the value predicted by the model and the real value. The average difference is equal to 21 injured people, which is approximately 2% of the total number of victims.

The verification of the correctness of the obtained model was carried out on the data set $S_{II}$. A fragment of the data that was used for the verification is presented in Table 3.

**Table 3.** Summary of parameters that characterize the developed mathematical model.

| Numeric Indicator | Unit | Voivodeship | | | |
|---|---|---|---|---|---|
| | | Dolnośląskie | Pomorskie | Małopolskie | Wielkopolskie |
| $V_1$ | Million PLN | 625.7 | 1589.6 | 924.3 | 919.9 |
| $P_1$ | Million PLN | 54.4 | 195.9 | 314.7 | 89.4 |
| $S_1$ | Number of entities | 4 | 7 | 0 | 1 |
| $E_1$ | Number of people | 7636 | 10,521 | 7864 | 10,916 |
| $EV_1$ | Number of people | 8397 | 14,353 | 8481 | 10,516 |
| ... | ... | ... | ... | ... | ... |
| $A$ | Number of people | 377 | 479 | 366 | 574 |

After conducting the calculations, the values of the multiple correlation coefficient $R_P$ and the coefficient of determination $R^2$ were again determined. The obtained values of the above statistics were, respectively, $R_P = 0.927$ and $R^2 = 0.860$, and prove a strong correlation of the empirical values with the values obtained using the mathematical model [38], and also a good quality of the model's adjustment to the analyzed phenomenon [39].

*4.2. Computational Example*

Using the developed multifactor mathematical model, research was conducted in order to determine the impact of a given sub-factor on the number of people injured in occupational accidents. For this purpose, the values of correlation coefficients between particular indicators, which are presented in Table 4, were determined.

**Table 4.** Values of correlation coefficients.

| Indicator $I_n$ | $V_1$ | $P_1$ | $P_2$ | $P_3$ | $P_4$ | $P_5$ | $S_1$ | $S_2$ | $S_3$ | $E_1$ | $E_2$ | $EV_1$ | $EV_2$ |
|---|---|---|---|---|---|---|---|---|---|---|---|---|---|
| $V_1$ | 1.00 | 0.21 | −0.41 | 0.34 | 0.13 | 0.37 | 0.41 | −0.16 | −0.46 | 0.53 | 0.41 | 0.68 | 0.55 |
| $P_1$ | 0.21 | 1.00 | 0.06 | 0.41 | 0.17 | 0.15 | 0.46 | 0.29 | 0.22 | 0.46 | 0.66 | 0.44 | 0.46 |
| $P_2$ | −0.41 | 0.06 | 1.00 | 0.02 | −0.31 | −0.10 | 0.01 | 0.08 | −0.37 | −0.61 | −0.40 | −0.31 | −0.08 |
| $P_3$ | 0.34 | 0.41 | 0.02 | 1.00 | 0.02 | 0.24 | 0.37 | 0.07 | 0.06 | 0.32 | 0.40 | 0.33 | 0.34 |
| $P_4$ | 0.13 | 0.17 | −0.31 | 0.02 | 1.00 | 0.49 | 0.12 | 0.13 | 0.18 | 0.34 | 0.36 | 0.19 | 0.22 |
| $P_5$ | 0.37 | 0.15 | −0.10 | 0.24 | 0.49 | 1.00 | 0.47 | 0.49 | −0.23 | 0.11 | 0.13 | 0.17 | 0.61 |
| $S_1$ | 0.41 | 0.46 | 0.01 | 0.37 | 0.12 | 0.47 | 1.00 | 0.40 | −0.13 | 0.42 | 0.57 | 0.55 | 0.73 |
| $S_2$ | −0.16 | 0.29 | 0.08 | 0.07 | 0.13 | 0.49 | 0.40 | 1.00 | 0.22 | 0.19 | 0.26 | 0.17 | 0.64 |
| $S_3$ | −0.46 | 0.22 | −0.37 | 0.06 | 0.18 | −0.23 | −0.13 | 0.22 | 1.00 | 0.38 | 0.39 | −0.06 | −0.19 |
| $E_1$ | 0.53 | 0.46 | −0.61 | 0.32 | 0.34 | 0.11 | 0.42 | 0.19 | 0.38 | 1.00 | 0.93 | 0.82 | 0.50 |
| $E_2$ | 0.41 | 0.66 | −0.40 | 0.40 | 0.36 | 0.13 | 0.57 | 0.26 | 0.39 | 0.93 | 1.00 | 0.82 | 0.56 |
| $EV_1$ | 0.68 | 0.44 | −0.31 | 0.33 | 0.19 | 0.17 | 0.55 | 0.17 | −0.06 | 0.82 | 0.82 | 1.00 | 0.72 |
| $EV_2$ | 0.55 | 0.46 | −0.08 | 0.34 | 0.22 | 0.61 | 0.73 | 0.64 | −0.19 | 0.50 | 0.56 | 0.72 | 1.00 |

$P_2$—the capital of enterprises, $P_3$—investment expenditure in the construction industry, $P_4$—investment expenditure for machines, technical devices and tools, $P_5$—investment expenditure for means of transport, $S_2$—the number of entities registered as commercial companies, $S_3$—the number of entities run by self-employed people, $E_2$—the number of owners and co-owners of enterprises, as well as family members who help free of charge, and $EV_2$—the number of employed people who have returned from parental leave.

The percentage change in the number of people injured in occupational accidents in the construction industry was then determined by the percentage change in the value of each indicator $I_n$. The degree of correlation dependence between the indicators in the model was considered in the calculations. The obtained results are shown in Table 5.

**Table 5.** The percentage change in the number of people injured in occupational accidents in the construction industry in relation to the percentage change in the value of the selected indicator.

| Change in the Value of Indicator $I_n$ | +10% | +20% | +30% | The Type of Impact on the Accident Rate |
|:---:|:---:|:---:|:---:|:---:|
| Indicator $I_n$ | Percentage Change [%] | | | Factor |
| $V_1$ | 4.80 | 9.59 | 14.39 | |
| $P_2$ | 5.58 | 11.15 | 16.73 | |
| $P_3$ | 3.41 | 6.81 | 10.22 | |
| $P_4$ | 6.93 | 13.87 | 20.80 | |
| $P_5$ | 7.68 | 15.35 | 23.03 | |
| $S_1$ | 1.78 | 3.55 | 5.33 | stimulating |
| $S_3$ | 10.35 | 20.71 | 31.06 | |
| $E_1$ | 8.67 | 17.33 | 26.00 | |
| $E_2$ | 11.93 | 23.85 | 35.78 | |
| $EV_1$ | 10.78 | 21.57 | 32.35 | |
| $EV_2$ | 4.75 | 9.50 | 14.24 | |
| $P_1$ | -6.60 | -13.20 | -19.81 | unstimulating |
| $S_2$ | -4.95 | -9.90 | -14.85 | |

The analysis of the obtained results shows that individual sub-factors influence the number of people injured in occupational accidents to a different extent.

Some of them have a stimulating character, and thus cause an increase in the number of people injured in occupational accidents, while others are unstimulating and have a negative impact on this number. Based on the figures in Table 5, the following indicators are of the greatest importance with regards to the increase in the number of people injured in occupational accidents in the construction industry:

$E_2$—the number of owners and co-owners of enterprises, as well as family members who help free of charge,
$EV_1$—the number of people employed, and
$S_3$—the number of entities run by self-employed people.

Among the factors included in the model, only two are beneficial for the improvement of occupational safety, and thus they unstimulate the accident rate. These are:

$S_2$—the number of entities registered as commercial companies, and
$P_1$—the value of current assets of construction enterprises associated with finished products.

Based on the numerical data included in Table 5, it can be stated that the structure of entities conducting economic activity, and also the phenomenon of staff turnover, is of the greatest importance with regards to the level of occupational safety in the construction industry. In Poland, construction enterprises operate as: commercial law companies, state-owned enterprises, civil partnerships, cooperatives, and self-employed people. The conducted calculations indicated that the increase of 10% in the number of enterprises with a self-employed status ($S_3$) causes an increase in the number of people injured in accidents by 10.35%.

Enterprises in this group are primarily micro, small- and medium-sized construction companies. Based on the analysis of statistical data, it is clear that the vast majority of accidents occur in these enterprises [36]. In addition, it is often the owners and their family members who work in such enterprises ($E_2$). An increase in indicator ($E_2$) by 10% causes an increase in the number of victims by 11.93%. The second factor that has a large impact on the accident rate is staff turnover. Conducted research showed that every year a significant number of people leave their current workplace and new employees are taken in their place. These employees do not have adequate professional experience

($EV_1$) [31]. The replacement of 10% of experienced employees with new people causes an increase in the number of people injured in accidents by 10.78%.

In turn, commercial law companies ($S_2$) have a positive impact on the level of occupational safety in the construction industry (these are the so-called accident rate destimulants). The increase in the number of such entities by 10% reduces the number of people injured in accidents by 4.95%. This observation also coincides with the results of statistical surveys, which confirm the low accident rate level in large enterprises, including those associated in the "Agreement for safety in the construction industry". Commercial law companies are affiliated with this agreement.

The second factor that unstimulates the number of people injured in occupational accidents is the value of the current assets of construction enterprises for finished products. Large stocks of finished products may have an impact on the reduction of the production rate and a reduction in the number of executed construction projects, which translates into a decrease in the number of occupational accidents. An increase of 10% in current assets that are attributable to finished products reduces the number of people injured in occupational accidents by 6.60%.

The proposed model and conducted analyses are of a theoretical nature and may be treated as approximate prognostic indicators. The real accident rate will also depend on many other factors that were not considered in the research, including the so-called hidden and unknown factors that have their sources in, among others, the field of psychology, sociology and culture.

## 5. Conclusions

This article proposes a methodology for assessing the impact of socio-economic factors on the accident rate in the construction industry. The following conclusions were made:

1. The multifactorial linear regression model, in which the independent variables are the selected indicators that characterize the construction production, and the dependent variable is the number of people injured in accidents, reflects the phenomenon of accidentality in the construction industry very well, as confirmed by conventional statistical indicators.
2. The analysis of the results obtained on the basis of the model indicates that particular socio-economic factors influence the number of people injured in occupational accidents to a different extent. This fact enables factors that stimulate and unstimulate accidents to be identified.
3. The stimulating factors that increase the number of people injured in occupational accidents include $V_1$, $P_2$, $P_3$, $P_4$, $P_5$, $S_1$, $E_2$, $E_1$, $E_2$, $EV_1$, and $EV_2$. The unstimulating factors that cause a decrease in the number of people injured in occupational accidents include $P_1$ and $S_2$.
4. The model proposed in the article can be used in the field of scientific research and also in engineering practice in the area of issues related to the management of occupational safety in the construction industry. The practical aspect of using the model and obtained results is connected to the possibility of drawing conclusions that can be the basis for insurance analyses and for the estimation of occupational risk in the construction industry.
5. The conducted tests and analyses contain the following limitations:

   - Only socio-economic indicators that can be presented in the form of numerical values were used to build the model.
   - Studies of published statistical data also indicate the impact of the personal characteristics of employees on the accident rate. These factors are directly related to the construction site and the construction process that is being carried out, which is why they were not included in these studies.
   - During the acquisition of data concerning the value of individual indicators that describe the situation of the Polish construction industry, it was determined that the methodology for collecting statistical data was modified in 2005. Therefore, previous numerical data could not be used in the research because it did not adhere to later data and thus could not create a

common set. In addition, the data sets from 2006 and 2007 are incomplete. Therefore, the set used in the research is left-bounded up to 2008.

- Because of the fact that individual voivodeships in Poland are characterized by different levels of economic development, the model was developed for a selected group of voivodeships. This is a significant limitation regarding its applicability. The model can only be used to predict the number of injured people in the selected four voivodeships. The use of a model to predict the number of people injured in other voivodeships is associated with a high probability that the obtained results will be more likely to differ from the real ones.

**Author Contributions:** Formal analysis, T.N.; Methodology, B.H. and T.N.; Project administration, B.H.; Software, T.N.; Supervision, B.H.

**Funding:** This research was funded by NCBiR within the framework of the Programme for Applied Research grant number PBS3/A2/19/2015.

**Acknowledgments:** The article is the result of the implementation by the authors of research project No. 244388 "Model of the assessment of risk of the occurrence of building catastrophes, accidents and dangerous events at workplaces with the use of scaffolding", financed by NCBiR within the framework of the Programme for Applied Research on the basis of contract No. PBS3/A2/19/2015.

**Conflicts of Interest:** The authors declare no conflict of interest.

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
