# Peer review of "Analysis of the Influence of Socio-Economic Factors on Occupational Safety in the Construction Industry"

_sustainability, doi:10.3390/su11164469_

Round 1

Reviewer 1 Report

The paper proposes an original approach for identifying factors determining occupational safety in the construction sector and well fits with the aims of  the Journal Sustainability. The organization of the paper is adequate but the references are not complete.   Some modifications are required to improve the overall quality of the paper.

Introduction is not adequately developed and some further should be added. A proper quantitative presentation of the problem of occupational accidents in the construction sector in Poland by adopting standard indicators regarding injuries and fatalities and the trend. This sector is one of the most hazardous and proper citations to long-term studies in different Countries seem advisable to highlight better why risk is important in the examined context and which kind of risks can be evidenced also compared to other sectors. At line 42, I would find relevant at least following studies to be referenced and commented:

Segarra Cañamares, M., Villena Escribano, B.M., González García, M.N., Romero Barriuso, A., Rodríguez Sáiz, A.  2017. Occupational risk-prevention diagnosis: A study of construction SMEs in Spain. Safety Science 92, 104-115.

Fabiano, B., Parentini, I., Ferraiolo, A., Pastorino, R. 1995. A century of accidents in the Italian industry - Relationship with the production cycle. Safety Science, 21, 65-74.

Gao, S., Low, S.P., Howe, H.J.A.  2018. Systemic lapses as the main causes of accidents in the Singapore construction industry. Civil Engineering and Environmental Systems 35, 81-98.

Literature survey This chapter is quite adequately structured even if, for the sake of completeness, I’d find relevant adding some recent approaches aiming at prevention and cause analysis by adequate modelling approaches. The problem of prevention is also faced by analysing  hazardous attitudes and near-misses and this aspect should be adequately evidenced. I suggest following sources of information to be included and commented in the state-of-the-art.

Sarkar, S., Raj, R., Vinay, S., Maiti, J., Pratihar, D.K. 2019. An optimization-based decision tree approach for predicting slip-trip-fall accidents at work Safety Science 118, 57-69.

Cermelli D., Pettinato M., Curro' F., Fabiano B., 2019, Major accident prevention: a construction site approach for pro-active management of unsafe conditions, Chemical Engineering Transactions, 74, 1387-1392 DOI:10.3303/CET1974232

Hoła, B., Szóstak, M. 2019.  Modeling of the accidentality phenomenon in the construction industry. Applied Sciences (Switzerland) 9,1878.

Proposed research methodology. This chapter is redundant and some of its content is rather trivial being siply referred to Statistica package. Authors should rewrite the chapter leaving only the original content of the approach.

Research results and their analysis.  I’d suggest relabeling this chapter as Results and discussion. The characterization of the sample is not given at all and should be added together with all pertinent quantitative details, even if already presented in other publications. At least an idea of the composition and extent of the sample and of its characteristics that are elaborated should be given as raw data. Here the discussion is rather simplified; the presentation of ?p, R2 , ?2s remains a simple statistical elaboration without discussing in detail how these indicators affect the reliability of the assessment. Eq. 2 is not a mathematical model but only a multiple regression equation obtained by the software. Line 227: The sentence “The abovementioned sub-factors were selected from a set of 104 elements using stepwise linear progressive regression.” should be adequately explained with all information and/or proper citations.

The safety performance related to occupational accidents regarding of the sample (possibly over different years) are missing. Several factors regarding workforce distribution well relevant to accident statistics are not considered at all. A justification for the assumptions of neglecting several workforce characteristics should be added.  Limitations of the proposed sample selection and possible improvements may be added as well.

The limitations of the study should be better stated in Abstract and in Conclusions.

 A final check of the English style is required.

Reviewer 2 Report

The study is well thought and designed.  The biggest issue is the verification or at least attempt at proof of concept.  The model was built right ( validated) through the statistical tests.  Was the right model built, i.e., was it verified?   One would expect an attempt to run a set of data from an independent project or projects and anticipate the outcomes and then compare to the known outcomes (MSE).  This would go along way to substantiate the claims made in the conclusion of the paper.  Otherwise, the paper is only a proposed model that is hypothetically sound but is not useful. 

Round 2

Reviewer 1 Report

Authors addressed the major part of comments by reviewers, thus increasing the quality of the paper. For better readability some minor revision is requested: Conclusions can be conveniently shortened (e.g. reducing first sentences and point 5, deleting point 4 (as it can be argued from point 3), etc.) and should be merged with the subsequent chapter 6 slightly reduced and relabelled as point 5 of conclusions.

In reducing Conclusions, I suggest to conclude line 405 as " ... accidentality in the construction industry very well, as confirmed by conventional statsitical indicators". , deleting the following lines 406-410.

Also the notations repeated in the point 3 of Conclusions can be omitted.

Author Response

Dear Reviewer,

Thank you very much for your review and critical comments, which will certainly improve the quality of the article. In the revised version of the article, we have included all the comments and we provide detailed answers below:

Point 1: “For better readability some minor revision is requested: Conclusions can be conveniently shortened (e.g. reducing first sentences and point 5, deleting point 4 (as it can be argued from point 3), etc.) and should be merged with the subsequent chapter 6 slightly reduced and relabelled as point 5 of conclusions.”

Response 1: Chapters 5 and 6 have been merged into one Chapter - Conclusion. All content has been reduced as directed.

Point 2: “In reducing Conclusions, I suggest to conclude line 405 as " ... accidentality in the construction industry very well, as confirmed by conventional statsitical indicators". , deleting the following lines 406-410.”

Response 2: The article shortened point 1 of the Conclusions introducing the recommended content.

Point 3: “Also the notations repeated in the point 3 of Conclusions can be omitted.”

Response 3: All factors mentioned and described in section 3 of the Conclusions have been shortened leaving only designations.

All changes made to the content of the article have been marked in red.

Best Regards,

Authors